# Association between Diet Quality and Stroke among Chinese Adults: Results from China Health and Nutrition Survey 2011

**DOI:** 10.3390/nu15143229

**Published:** 2023-07-20

**Authors:** Disi Gao, Huan Wang, Yue Wang, Sheng Ma, Zhiyong Zou

**Affiliations:** 1School of Health Humanities, Peking University, Beijing 100191, China; gaodisi@bjmu.edu.cn (D.G.); wangyues@bjmu.edu.cn (Y.W.); 2Institute of Child and Adolescent Health, School of Public Health, Peking University, Beijing 100191, China; 2211110224@bjmu.edu.cn (H.W.); 2111210073@bjmu.edu.cn (S.M.)

**Keywords:** diet quality, global dietary recommendations, stroke, Chinese, women

## Abstract

The low-burden Diet Quality Questionnaire (DQQ) is a standardized tool to collect indicators of dietary adequacy as well as indicators of the protection of health against noncommunicable diseases (NCDs) within the framework of the global diet quality project. Stroke is the leading cause of the cardiovascular disease burden in China, with poor diet being one of the major risk factors. In this study, we aimed to understand the association of several indicators of diet quality derived from the DQQ with stroke among Chinese adults and, further, to examine the gender differences using the 2011 wave of the China Health and Nutrition Survey. Multivariable logistic regression was used to examine the associations of the NCD-Protect score, NCD-Risk score, and global dietary recommendations score (GDR) score with stroke. There were 192 stroke cases (121 in men and 71 in women) of 12,051 adults. The continuous NCD-Risk score was positively associated with stroke in women (odds ratio (OR) = 1.52, 95% confidence interval (CI): 1.13–2.06). When compared with women with an NCD-Risk score of 0 points, those with an NCD-Risk score ≥2 points had a higher risk of stroke (OR = 2.71, 95% CI: 1.35–5.43). In addition, compared with women with a GDR score ≤0, those with a GDR score ≥2 points had lower odds of stroke (OR = 0.42, 95% CI: 0.22–0.77). Poor diet quality, as reflected by the NCD-Risk score, was associated with an increased risk of stroke in Chinese women, but not in men. Our findings provided evidence that an optimal diet quality could be conducive to preventing stroke for Chinese women and suggested a diverse diet characterized by the limited consumption of unhealthy foods, such as red meat, processed meat, sweets, soft drinks (sodas), and packaged ultra-processed salty snacks.

## 1. Introduction

Stroke is the leading cause of mortality and long-term disability worldwide, resulting in a substantial economic burden on individuals, families, and society [1,2]. Globally, the lifetime risk of stroke is higher among women than men [3]. With the growth of the aging population, the prevalence of stroke is projected to increase, particularly among elderly women [4,5]. According to the Global Burden of Disease Study (GBD), in 2019, estimated 2.19 million deaths due to stroke in China, with an increment of 32.3% in the mortality rate in the past three decades [6]. 

Optimal diet, smoking cessation, and blood pressure control would be the most important interventions to prevent stroke [7]. In China, the leading risk factors for stroke among women were high systolic blood pressure, ambient particulate matter pollution, high body mass index (BMI), and a diet high in sodium [6].

Proper diet and nutrition are important to prevent stroke, and internationally accepted dietary guidelines are available for stroke prevention [8]. The American College of Cardiology/American Heart Association (ACC/AHA) recommends a diet high in fruits, vegetables, and whole grains and the restricted intake of red meat, sweets, and carbonated beverages [9]. The guidelines of the European Society of Cardiology for a healthy diet make similar recommendations, which advise the limit of salt intake to <5 g/day, with 200 g of fruits and vegetables each per day and two servings of oily fish per week [10]. Several observational and cohort studies also revealed that diet plays an independent role in stroke incidence [11]. Both the Prevención con Dieta Mediterránea (PREDIMED) study and the REasons for Geographic and Racial Differences in Stroke (REGARDS) cohort have demonstrated a lower risk of ischemic stroke with the adherence to the Mediterranean diet [12,13], among which cruciferous and green leafy vegetables and citrus fruits and juices have the highest protective value; each additional one serving per day was associated with a 6% lower risk of ischemic stroke [14]. Recent surveys have indicated a lower risk of stroke with two or four or more than five servings of fish per week [15], and extra virgin olive in the context of the Mediterranean diet has also been proven to reduce effectively the risk of atrial fibrillation [13]. In addition, meta-analyses have shown that red meat, especially processed meat, consumption is associated with an increased incidence of stroke [16], and higher salt intake is associated with a greater risk of fatal and nonfatal stroke [17]. 

Food types and a dietetic history are two aspects of understanding diet, as they allow one to better analyze the results of nutritional studies [8]. Recently, a low-burden Diet Quality Questionnaire (DQQ) based on the framework of global diet quality has been developed to collect dietary data [18]. The DQQ is a standardized tool that can be used to construct several diet quality indicators at the population level, such as zero vegetable or fruit consumption or sugar-sweetened soft drink consumption [19]. By using “yes/no” questions about foods or drinks, respondents can easily understand the survey, which can be administered in only approx. 5 min [20]. The DQQ survey also collects data regarding indicators of the protection of health against noncommunicable diseases (NCDs), including NCD-Protect, NCD-Risk, and the global dietary recommendations score (GDR). Thus, we would expect that the diet quality scores derived from the DQQ may be associated with stroke, because it captures the consumption of food groups meeting the ACC/AHA recommendations [9]. In the present study, we aimed to examine the association between the diet quality scores and stroke and to identify gender differences in stroke incidence in Chinese adults.

## 2. Materials and Methods

### 2.1. Data Resources and Study Participants

Data in this study were obtained from the China Health and Nutrition Survey (CHNS), which is an ongoing and international collaborative project between the Carolina Population Center at the University of North Carolina at Chapel Hill and the National Institute for Nutrition and Health (NINH) at the Chinese Center for Disease Control and Prevention (CCDC). The CHNS aims to understand the interplay of socio-economic transition and nutrition and health-related outcomes in China [21]. This project was reviewed and approved by the corresponding institutional review committees (2015017) [22]. Further details of the CHNS data can be obtained from the cohort profile [22].

There were 15,725 participants in the 2011 wave of the CHNS, and we included all adults aged 18 years or older (*n* = 13,097). After further exclusion of 974 patients without complete records on diet or anthropometrics (including 70 without data on stroke, 229 without data on diet quality questionnaire, and 675 without data on demographic, behavioral, or physical factors) and 72 with implausible dietary intakes, a total of 12,051 adults with complete data were included in this cross-sectional study (Figure 1).

### 2.2. Stroke and Other Variables

Body mass index (BMI) was calculated as weight (kg) divided by height squared (m^2^), and waist circumference (WC, accurate to 0.1 cm) was measured at the midpoint between the lowest rib and the iliac crest using non-elastic tapes in the standing position after normal expiration [23]. Overweight, general obesity, and abdominal obesity were defined according to the criteria of weight for Chinese adults [24]. Both systolic blood pressure (SBP) and diastolic blood pressure (DBP) were measured with mercury sphygmomanometer according to a standard protocol through standard mercury [25]. In this survey, information on stroke was collected based on self-reported disease history, and the relevant question was “Diagnosed with Apoplexy?”.

### 2.3. Dietary Data Collection and Assessment

The quantitative dietary data were collected using 24 h dietary recall by trained investigators from local Centers for Disease Control and Prevention for three consecutive days [26]. The reliability of the 24 h recall has been validated by previous research, and further details on the dietary interview have been reported elsewhere [27].

The Diet Quality Questionnaire (DQQ) is developed to assess rapidly the diet quality at the population level globally [18,22], and the DQQ for China is a validated tool to capture more than 95% of the population who consumed sentinel foods in each of 29 food groups [19]. In this study, dietary intake of foods in the first day during the three consecutive days was coded into 29 food groups according to the DQQ for China [19]. Further information on the DQQ has been previously published [19].

The NCD-Protect score, NCD-Risk score, and GDR score form a suite of low-burden diet quality indicators derived from the DQQ that can be calculated simply, which can feasibly monitor the diet quality holistically and reflect diet quality relevant to policies and programs [18]. The NCD-Protect score, NCD-Risk score, and GDR score were constructed from the dietary intake data: (1) The NCD-Protect score reflects 5 global recommendations on nutritious foods for healthy diets, including fruits and vegetables, whole grains, legumes, nuts, and seeds. (2) The NCD-Risk score reflects 6 global recommendations on dietary components to limit certain foods, including processed meat, red meat, and other food groups that are high in sugar, salt, total fat, or saturated fat. (3) The GDR score is a score reflecting all 11 global recommendations, calculated as NCD-Protect score minus NCD-Risk score. The NCD-Protect score and NCD-Risk score ranged from 0 to 9 points, and the GDR score ranged from −9 to 9 points. A lower GDR score, lower NCD-Protect score, and a higher NCD-Risk score indicate a poorer diet quality [18,28]. 

### 2.4. Statistical Analysis

Data were expressed as numbers (percentage) for categorical variables and as medians ± interquartile ranges (IQR) or means ± standard deviations (SD) for continuous variables (age, GDR score, BMI, and WC). The Chi-square test, Wilcoxon rank test, and *t*-test were used to compare differences in characteristics between men and women. The multivariable logistic regression analyses were used to evaluate the associations of the diet quality scores with stroke, and odds ratios (OR) and 95% confidence intervals (CI) were estimated after adjusting for age, marital status, residence, smoking status, drinking status, educational level, BMI, SBP, and total energy (total population also adjusted for sex). A two-sided *p*-value < 0.05 was considered as statistically significant. All analyses were performed using SAS 9.4 (SAS Institute, Cary, NC, USA).

## 3. Results

### 3.1. Participant Characteristics

A total of 12,051 participants (men: 46.7%) with a median age of 51 years were included in this study, with 29.2% of them aged 60 years or older, and 42.6% lived in the urban areas. Most of them were married (84.6%), and more than a half (64.0%) had middle school or higher education. Approximately one-third of participants smoked (30.5%) or drank (33.5%), and men were more likely to smoke and drink than women (*p* < 0.01). The average BMI was 23.9 ± 4.6 kg/m^2^, and 44.8% of participants were overweight or obese, 37.5% had abdominal obesity, and 29.0% had suffered hypertension. Men had significantly higher levels of WC, SBP, and DBP and a higher prevalence of hypertension than women (*p* < 0.01). The GDR score was also significantly different between men and women (*p* < 0.01). Men also consumed more energy, carbohydrates, protein, fat, calcium, and sodium per day than women (*p* < 0.01; Table 1).

### 3.2. Distribution of Diet Quality Scores and Stroke

The prevalence of stroke was 1.6% in the entire population, with 2.2% in men and 1.1% in women. Table 2 also shows the prevalence of stroke across the diet quality scores by sex. Among men with an NCD-Protect score of less than 2 points, the prevalence of stroke was 1.9%; among women with an NCD-Risk score of more than 2 points, the prevalence of stroke was 1.9% (Table 2).

### 3.3. Associations of Diet Quality Scores with Stroke

After adjustment for age, marital status, residence, smoking status, drinking status, educational level, BMI, SBP, and total energy, the continuous NCD-Risk score was positively associated with stroke in women (OR = 1.52, 95% CI: 1.13–2.06), especially in those women with an NCD-Risk score ≥2 points (OR = 2.71, 95% CI: 1.35–5.43). As for the categorical diet quality scores, compared with women with a GDR score ≤0, those with a GDR score ≥2 points had a lower odds of stroke (OR = 0.42, 95% CI: 0.22–0.77; Table 3).

Stratified analyses by residence (rural vs. urban), age (18–59 vs. 60 years or older) are shown in Figure 2, with similar positive associations between the NCD-Risk score ≥2 points and stroke among elderly people, especially those women who were 60 years or older.

Logistic regression analyses were used to calculate OR and 95% CI with adjustment for age, marital status, residence, smoking status, drinking status, educational level, body mass index, systolic blood pressure, and total energy (total population also adjusted for sex).

## 4. Discussion

In this national analysis of 12,051 adults from the 2011 wave of the CHNS, we observed that the NCD-Risk score was positively associated with the odds of stroke among Chinese women; when the NCD-Risk score ≥ 2 points, women (especially those 60 years or older) would have a higher odds of stroke. The present study suggests that the poor diet quality reflected by the NCD-Risk score is associated with a higher risk of stroke among Chinese women. 

This study used the NCD-Risk score, NCD-Protect score, and GDR score from the China-adapted DQQ to assess diet quality, which reflects adherence to the Global Dietary Recommendations [18]. We observed higher odds of stroke among women with a higher NCD-Risk score, especially for elderly women (aged 60 years or older), which was supported by other studies [29]. Women face a disproportionate burden of stroke mortality and disability [30], and the stroke risk factors, incidence, and death increase from around the time of menopause [31]. A multicenter longitudinal study of 93,676 postmenopausal women aged 50 to 79 years demonstrated that the risk of stroke increased along with the increasing consumption of artificial sweeteners [32]. Another study showed that meat (especially red meat) contains carnitine, and egg yolks contain phosphatidylcholine, both of which can be converted by the intestinal microbiome to trimethylamine and then oxidized in the liver to trimethlylaminen-oxide (TMAO). In addition, TMAO has been proved to cause atherosclerosis in animal models, thereby affecting the stoke incidence [33]. Red meat and foods high in sugar, salt, total fat, or saturated fat have been documented to be dietary risk factors for stroke [34]. In the previous published guidelines and scoping reviews [29,35]—especially the guidelines for the prevention of stroke in women—diabetes, hypertension, and obesity have been listed as risk factors for stroke, and these diseases were related significantly to the above foods [36]. In China, the major cause of stroke is hypertension, which is often caused by the excessive intake of sodium salts [37,38]. This evidence supports the view that poor diet quality determined by the NCD-Risk score is associated with an increased risk of stroke in women.

In this study, we observed an inverse association between the GDR score ≥2 points and stroke in women; this finding implies that the more food groups women eat, the lower the incidence of stroke they may have. The GDR score was aligned with the 11 global dietary recommendations, and the World Health Organization (Geneva, Switzerland) proposes these general recommendations based on evidence related to diet-related NCD risks, such as cardiovascular disease, including stroke [39,40]. Although no significant association was found between diet quality and stroke among men, several recent studies have reported the stroke prevention effects of specific nutrients or dietary patterns to support the idea that diet, as a modifiable environmental factor, can efficiently reduce the risk of stroke [41,42]. As part of the Mediterranean diet, fish consumption has been proven to be associated with a decreased risk of stroke [43], and the NHANES I Study found that in women aged 45 to 74 years who consumed fish more than once a week, the age-adjusted risk of stroke incidence was reduced by half compared with women who never ate fish [44]; the explanation for these results likely resides in the findings that the consumption of some fish is correlated with plasma levels of long-chainω-3 fatty acids [45]. Similarly, fruits and vegetables were also essential components of the Mediterranean diet, which protects effectively against ischemic and hemorrhagic stroke. A systematic review of 95 epidemiological studies has shown that moderate consumption (3–4 servings per day) of fruits, vegetables, and legumes was associated with a reduced risk of 27% to 39% of stroke compared with less than 1 serving/day [46]. The Dietary Approaches to Stop Hypertension (DASH) diet also recommends increasing the intake of potassium from the consumption of fruits, vegetables, nuts, and whole grains in order to lower blood pressure to decrease the risk of stroke [47]. A meta-analysis of 12 prospective cohort studies concluded that the DASH diet is associated with a decreased risk of stroke [48]. In addition, the consumption of cereal grains to prevent stroke has been recommended by the American Heart Association/American Stroke Association and recorded in their comprehensive guidelines on the dietary factors for stroke prevention [49]. A number of prospective epidemiological cohort studies also showed a reduction in stroke risk with a higher intake of flavonols and flavones [50,51]. It is also clear that B vitamins could lower homocysteine to reduce the risk of stroke [34]. Different foods and food groups were excellent sources of various macronutrients and micronutrients [52]; therefore, from the public health viewpoint, the optimal health benefits with a reduction in stroke incidence can be achieved by a diversified diet.

This is the first study to assess the association between the diet quality scores and stroke in Chinese adults. Several limitations of this study should be noted. Firstly, we obtained data from only the 2011 wave of the CHNS. This cross-sectional study lacks a chronological sequence and cannot prove a causal relationship between diet and stroke; therefore, further studies are needed to ascertain the mechanism underlying the association between diet quality and stroke incidence. Secondly, although factors such as age, educational level, and total energy were accounted for in our analyses, residual confounding factors cannot be ruled out, such as taking contraceptives, physical inactivity, or basic diseases. Our study did not find the relationship between diet and incidence of stroke in men, which is also one of the limitations of this study. There were many reasons that lead to this result, but the fundamental reason may be that women were more likely to suffer from stroke than men. The Framingham cohort has shown that women have a higher lifetime risk of stroke than men [53], and for a 55-year-old individual, stroke is more likely to be the first manifestation of cardiovascular disease in women [54]. There are also some suggestions that women have a higher incidence of stroke than men in the oldest category (>85 years) [55,56].

## 5. Conclusions

In conclusion, the NCD-Risk score is associated with an increased risk of stroke among Chinese women. A diverse diet characterized by the limited consumption of unhealthy foods is suggested to be conducive to the prevention of stroke.

## Figures and Tables

**Figure 1 nutrients-15-03229-f001:**
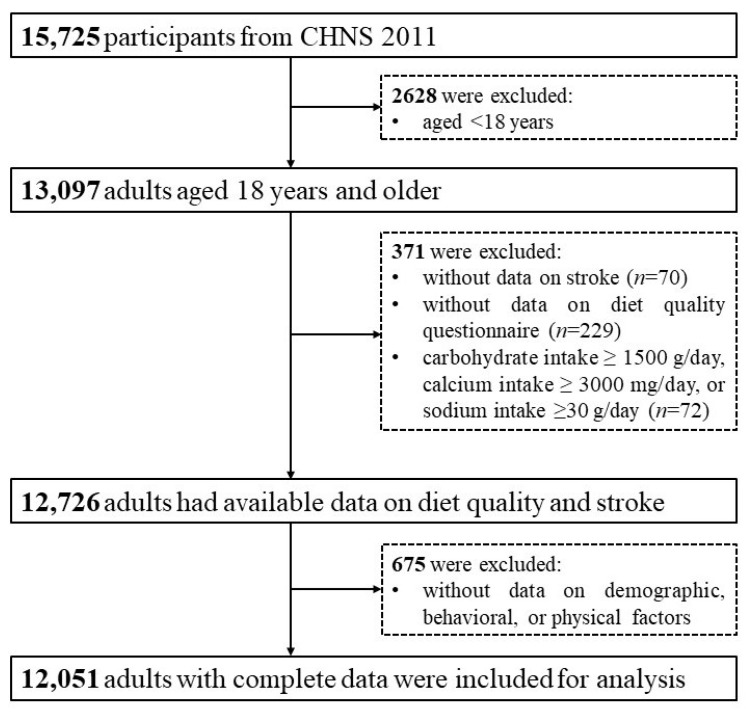
Flow chart for included participants.

**Figure 2 nutrients-15-03229-f002:**
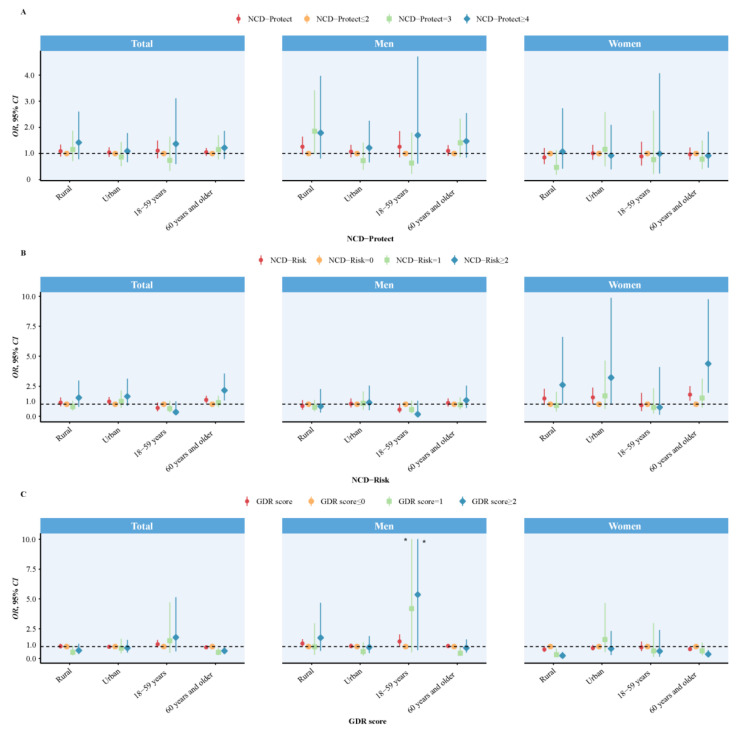
Association between diet quality scores and stroke, by age and residence. (**A**) shows the relationship between NCD-Protect scores and stroke, (**B**) shows the relationship between NCD-Risk scores and stroke, (**C**) shows the relationship between GDR scores and stroke. * The upper boundaries of 95% CI were excessively high (33.90 and 40.87), so they were plotted as 10 on the plot for visual comparison.

**Table 1 nutrients-15-03229-t001:** Characteristics of participants, categorized by sex.

Characteristics	Total (*n* = 12,051)	Men (*n* = 5628)	Women (*n* = 6423)	*p*-Value
Socio-demographic factors				
Age, median (IQR), years	51.00 (21.00)	52.00 (20.00)	51.00 (21.00)	0.076
18–59 years, *n* (%)	8528 (70.77)	3944 (70.08)	4584 (71.37)	0.120
60 years and older, *n* (%)	3523 (29.23)	1684 (29.92)	1839 (28.63)	-
Residence, *n* (%)				0.793
Rural	6916 (57.39)	3237 (57.52)	3679 (57.28)	
Urban	5135 (42.61)	2391 (42.48)	2744 (42.72)	
Marital status, *n* (%)				<0.001
Unmarried	659 (5.47)	388 (6.89)	271 (4.22)	
Married	10,191 (84.57)	4893 (86.94)	5298 (82.48)	
Divorced/widowed/separated	1201 (9.97)	347 (6.17)	854 (13.30)	
Educational levels, *n* (%)				<0.001
Primary school and lower	4344 (36.05)	1625 (28.87)	2719 (42.33)	
Middle school and higher	7707 (63.95)	4003 (71.13)	3704 (57.67)	
Behavioral/physical factors				
Smoking status, *n* (%)				<0.001
Never smoked	8373 (69.48)	2170 (38.56)	6203 (96.57)	
Smokes	3678 (30.52)	3458 (61.44)	220 (3.43)	
Drinking status, *n* (%)				<0.001
Never drank	8019 (66.54)	2309 (41.03)	5710 (88.90)	
Drinks	4032 (33.46)	3319 (58.97)	713 (11.10)	
BMI, mean (SD), kg/m^2^	23.94 (4.64)	24.01 (4.76)	23.87 (4.53)	0.102
Overweight/obesity, *n* (%)	5394 (44.76)	2590 (46.02)	2804 (43.66)	0.001
WC, mean (SD), cm	83.93 (11.11)	86.05 (10.84)	82.07 (11.01)	<0.001
Abdominal obesity, *n* (%)	4519 (37.54)	2062 (36.66)	2457 (38.32)	0.060
SBP, mean (SD), mmHg	124.60 (17.8)	126.36 (16.45)	123.06 (18.77)	<0.001
DBP, mean (SD), mmHg	79.32 (10.71)	80.93 (10.51)	77.91 (10.69)	<0.001
Hypertension, *n* (%)	3496 (29.01)	1774 (31.52)	1722 (26.81)	<0.001
Dietary factors				
GDR score, median (IQR)	2.00 (2.00)	2.00 (1.00)	2.00 (2.00)	<0.001
Total energy, median (IQR), kcal/day	1814.07 (944.09)	2007.45 (1018.42)	1676.19 (821.58)	<0.001
Carbohydrate, median (IQR), g/day	253.31 (158.33)	275.92 (174.09)	236.38 (141.85)	<0.001
Protein, median (IQR), g/day	62.42 (37.80)	68.67 (39.86)	57.55 (34.48)	<0.001
Fat, median (IQR), g/day	58.23 (51.03)	63.03 (53.79)	53.43 (48.10)	<0.001
Calcium, median (IQR), mg/day	360.88 (300.84)	382.18 (314.09)	342.69 (287.21)	<0.001
Sodium, median (IQR), mg/day	3725.38 (2569.57)	4002.53 (2738.79)	3523.83 (2354.39)	<0.001

**Table 2 nutrients-15-03229-t002:** Number and percent (%) of stroke cases by categories of diet quality scores.

	Total	Men	Women
Total stroke cases	192 (1.59)	121 (2.15)	71 (1.11)
NCD-Protect score			
≤2	85 (1.54)	50 (1.87)	35 (1.23)
3	57 (1.46)	38 (2.07)	19 (0.92)
≥4	50 (1.90)	33 (2.95)	17 (1.13)
NCD-Risk score			
0	56 (1.69)	39 (2.68)	17 (0.92)
1	95 (1.43)	62 (1.96)	33 (0.95)
≥2	41 (1.95)	20 (1.99)	21 (1.91)
GDR score			
≤0	33 (1.90)	17 (1.92)	16 (1.87)
1	46 (1.38)	24 (1.48)	22 (1.29)
≥2	113 (1.62)	80 (2.57)	33 (0.85)

**Table 3 nutrients-15-03229-t003:** Association between diet quality scores and stroke.

Diet Quality Scores	Total		Men		Women	
	OR (95% CI)	*p*-Value	OR (95% CI)	*p*-Value	OR (95% CI)	*p*-Value
NCD-Protect score						
Continuous	1.06 (0.93–1.20)	0.411	1.13 (0.95–1.33)	0.159	0.95 (0.77–1.18)	0.644
Categories						
≤2	1.00 (reference)	-	1.00 (reference)	-	1.00 (reference)	-
3	1.01 (0.71–1.43)	0.957	1.18 (0.75–1.83)	0.476	0.75 (0.42–1.34)	0.337
≥4	1.21 (0.83–1.76)	0.332	1.44 (0.89–2.33)	0.140	0.91 (0.49–1.68)	0.758
NCD-Risk score						
Continuous	1.17 (0.95–1.43)	0.132	0.96 (0.73–1.26)	0.760	1.52 (1.13–2.06)	0.006
Categories						
0	1.00 (reference)	-	1.00 (reference)	-	1.00 (reference)	-
1	0.97 (0.68–1.37)	0.848	0.87 (0.56–1.33)	0.513	1.17 (0.64–2.15)	0.616
≥2	1.47 (0.94–2.29)	0.089	0.95 (0.53–1.73)	0.872	2.71 (1.35–5.43)	0.005
GDR score						
Continuous	0.99 (0.89–1.12)	0.913	1.11 (0.96–1.29)	0.164	0.83 (0.69–1.00)	0.052
Categories						
≤0	1.00 (reference)	-	1.00 (reference)	-	1.00 (reference)	-
1	0.68 (0.42–1.08)	0.101	0.69 (0.36–1.32)	0.264	0.65 (0.33–1.28)	0.215
≥2	0.78 (0.52–1.17)	0.233	1.16 (0.67–2.01)	0.604	0.42 (0.22–0.77)	0.006

## Data Availability

The dataset in the present study are open-access and can be freely obtained from the CHNS website: https://www.cpc.unc.edu/projects/china/data/datasets/data_downloads/longitudinal (accessed on 16 May 2023).

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
