# Peer review of "Association between Diet Quality and Stroke among Chinese Adults: Results from China Health and Nutrition Survey 2011"

_nutrients, 2023, doi:10.3390/nu15143229_

Round 1

Reviewer 1 Report

Introduction

The authors conduct a novel study as according to them, it is the first study.

that evaluates the association between diet quality scores and stroke in Chinese adults.

The aim of the study is to analyze the association of various diet quality indicators derived from the low-burden Diet Quality Questionnaire (DQQ), with stroke among Chinese adults, and to assess gender differences.

Methodology

It is a cross-sectional study, and the experimental design is appropriate for testing the hypothesis that associates diet quality with stroke risk. The database they used is from the 2011 China Health and Nutrition Survey, an ongoing international collaborative project between the People's Center at the University of North Carolina at Chapel Hill and the Nutrition and Health Institute (NINH) at the Chinese Center for Disease Control and Prevention.

The inclusion/exclusion scheme of the 2011 CHNS participants, is well explained.

Results

The results of the manuscript can be reproduced by other authors based on the methodology used, as long as they are adjusted to the target population to be studied (e.g., definition of general obesity and abdominal obesity, which in this study are adapted to the Chinese population).  The statistical analysis is adequate.  The multivariate logistic regression, allows to know the predictive effect of different quantitative variables, in this study, the scores of different tools that evaluate the quality of the diet, according to gender, and other qualitative variables such as age group, and place of residence and the risk of stroke and the Odd ratio. 

The results presented in tables and/or figures are easy to interpret, as they indicate the overall figures, and also differentiated by gender. 

The statistical analysis is adequate since multivariate logistic regression allows us to know the predictive effect of different quantitative variables, in this study, the scores of different tools that evaluate the quality of the diet, according to gender, and other qualitative variables such as age group and place of residence.

The associations between diet quality scores, according to the questionnaires used, and stroke risk have been adjusted for other variables (age, marital status, residence, smoking, alcohol consumption, educational level, BMI, SBP, and total energy) to reduce the margin of error, although the authors explain that there may be residual confounding factors.

The figures and/or graphs are easy to understand throughout the manuscript. 

Discussion

Because it is a cross-sectional study, so a causal relationship between diet and stroke cannot be proven, so it is recommended that the authors continue to pursue this line of research in order to identify the underlying mechanisms between diet quality and the incidence of stroke, and its higher incidence in Chinese women.

The study seems interesting because modification of dietary habits is the main treatment for this disease and the recommendations derived from it may have an important future effect on the nutritional recommendations aimed at the Chinese population. 

The discussion of the study is well argued, and adequately interprets the results, comparing them with those of other published studies, and which coincides with the findings of other authors from different countries, with which it coincides in terms of the association between diet quality and the risk of cardiovascular disease, especially stroke.  Likewise, the authors have pointed out the possible limitations of the study (transversality, lack of chronological sequence, and failure to prove the causal relationship between diet and stroke).  Although the analyses took into account age, marital status, place of residence, smoking, alcohol consumption, educational level, BMI, systolic blood pressure (SBP) and total energy intake, they do not rule out that there may be residual confounding factors, such as contraceptive use, physical inactivity and underlying diseases:

The conclusions correspond to the objective of the study.

The study complied with institutional ethical review standards and had the informed consent of the participants.

Bibliographic references

Most of the bibliographic references are recent (last 5 years), while other older references are relevant to the subject.  There are not excessive self-citations

The manuscript is clear, coherent, well structured and well-founded. It can be a valuable contribution to implement changes in clinical and public health practice aimed at the Chinese population in particular.

The manuscript is clear, coherent, well structured and well-founded. It can be a valuable contribution to implement changes in clinical and public health practice aimed at the Chinese population in particular.

Reviewer 2 Report

I congratulate with the authors for the interesting paper.

Comments:

Line 96 reports the following text: " There were 15,275 participants in the 2011 wave of the CHNS.”, but the "Figure 1. Flow chart for included participants" (Line 104) reports 15725 participants from CHNS  2011. I suggests to verify it.

Lines 192-195 the following text: “Logistic regression analyses are used to calculate OR and 95% CI with adjustment for 192 age, marital status, residence, smoking status, drinking status, educational level, body 193 mass index, systolic blood pressure, and total energy (total population also adjusted for 194 sex)”, also, in lines 198-201 is reported,  I suggest to erase the repeated text.

In Line 197 is reported the Figure 2. I suggest to comment the results of the Figure 2 and to add the comments in the discussion.
